# Preservation of Vocal Function in Amyotrophic Lateral Sclerosis (ALS) Patients Following Percutaneous Dilatational Tracheostomy (PDT) and Adjuvant Therapies

**DOI:** 10.3390/biomedicines12081734

**Published:** 2024-08-02

**Authors:** Jae-Kook Yoo, Soon-Hee Kwon, Sul-Hee Yoon, Jeong-Eun Lee, Jong-Eun Jeon, Je-Hyuk Chung, Sang-Yoon Lee

**Affiliations:** 1Department of Neurolgy, The Rodem Hospital, Incheon 22142, Republic of Korea; koreadr@gmail.com (J.-K.Y.); rodem3@therodem.com (J.-E.J.); 2Department of Internal Medicine, The Rodem Hospital, Incheon 22142, Republic of Korea; rodem4@therodem.com; 3Department of Rehabilitation Medicine, The Rodem Hospital, Incheon 22142, Republic of Korea; ipraying@hanmail.net (J.-E.L.); rodem7@therodem.com (S.-Y.L.)

**Keywords:** percutaneous dilatational tracheostomy (PDT), placenta extract injection therapy, low-frequency electrical stimulation, respiratory rehabilitation, swallowing rehabilitation therapy

## Abstract

The study aimed to evaluate the efficacy of percutaneous dilatational tracheostomy (PDT) combined with adjuvant therapies in preserving vocal function in amyotrophic lateral sclerosis (ALS) patients. Methods: We performed a retrospective analysis of 47 ALS patients who underwent PDT at the Rodem Hospital from 2021 to 2023. Post-operatively, these patients were provided with a comprehensive treatment plan that included regenerative injection therapy, low-frequency electrical stimulation, respiratory rehabilitation, and swallowing rehabilitation therapy. Additionally, a balloon reduction program was implemented for effective tracheostomy tube (T-tube) management. The preservation of vocal functions was evaluated 4 weeks following the procedure. Results: While some patients maintained or slightly improved their ALSFRS-R speech scores, the overall trend indicated a decrease in speech scores post-PDT. This suggests that PDT in combination with adjuvant therapies may not universally improve vocal function, but can help maintain it in certain cases. Conclusions: Our findings indicate that PDT combined with mesotherapy, low-frequency electrical stimulation, and swallowing rehabilitation therapy may play a role in maintaining vocal function in limb type ALS patients, though further research is needed to optimize patient management and to validate these results.

## 1. Introduction

Amyotrophic lateral sclerosis (ALS) is a progressive neurodegenerative disorder affecting motor neurons in the brain and spinal cord, leading to muscle weakness and eventual respiratory failure [1]. As respiratory failure worsens, various non-invasive ventilation (NIV) techniques are used to manage symptoms. These include continuous positive airway pressure (CPAP), bilevel positive airway pressure (BiPAP), volume-assured synchronized intermittent mandatory ventilation (V-SIMV), volume-assured pressure support (V-AC), pressure-assured synchronized intermittent mandatory ventilation (P-SIMV), and pressure-assured pressure support (P-AC) modes [2]. Respiratory rehabilitation therapies, including the use of cough machines and vests, are also implemented [3]. When NIV proves insufficient or patient compliance declines, percutaneous dilatational tracheostomy (PDT) may be performed to manage respiratory failure [4].

ALS presents with diverse pathogenic mechanisms, necessitating various treatment approaches that have shown meaningful progress [5,6,7,8,9]. These include pharmacological interventions, gene therapy, and neuroprotective strategies aimed at slowing disease progression and alleviating symptoms. However, to improve vocal function, which significantly impacts the patients’ quality of life, direct interventions targeting the relevant anatomical and physiological aspects are essential. This study evaluates the impact of PDT when combined with adjuvant therapies on vocal function preservation in ALS patients.

The current literature [10,11,12] suggests that PDT combined with adjuvant therapies can play a crucial role in preserving vocal function, a key factor in maintaining quality of life for ALS patients. However, detailed studies evaluating the effectiveness of such combined treatments are limited. This study aims to fill this gap by assessing the impact of PDT and various adjuvant therapies on vocal function preservation in ALS patients. The adjuvant therapies investigated include regenerative injection therapy, low-frequency electrical stimulation, respiratory rehabilitation, and swallowing rehabilitation therapy.

This study seeks to build on these insights by evaluating the efficacy of PDT when combined with adjuvant therapies in preserving vocal function in ALS patients. By integrating findings from recent literature, this research aims to provide a comprehensive understanding of how these combined treatments can enhance patient outcomes and quality of life. Oxidative stress and inflammation are significant in neurodegenerative diseases, including ALS [7]. Adjuvant therapies such as antioxidants can mitigate oxidative damage and improve cellular function, potentially enhancing ALS treatment outcomes [8]. Cellular stress responses, such as autophagy and apoptosis, also play a role in neurodegeneration [7]. Genetic and environmental factors influence neurodegenerative diseases, and understanding these interactions can lead to personalized ALS treatments [6]. Advances in molecular techniques and bioinformatics have identified biomarkers and therapeutic targets that can improve the efficacy of interventions like PDT with adjuvant therapies [5].

This study builds on these insights by assessing PDT when combined with adjuvant therapies for vocal function preservation in ALS patients, aiming to enhance patient outcomes and quality of life.

## 2. Materials and Methods

We conducted a retrospective analysis of the medical records of 47 ALS patients who underwent Ultrasound (MedicalSK, Daegu, Republic of Korea, SONON 300L)-Guided PDT at Rodem Hospital between 2020 and 2023 (Table 1). Of these patients, 11 were diagnosed with bulbar type ALS, 2 with respiration type ALS and 34 with limb type ALS. The inclusion criteria were a diagnosis of limb type ALS based on the revised El Escorial criteria and the need for tracheostomy due to worsening respiratory function and low efficacy of NIV (Philips Trilogy 100, Philips Trilogy 300, Resmed Astral 150) care. The exclusion criteria included patients with bulbar and respiratory onset who had completely lost their voice.

Speech evaluation was based on a retrospective evaluation of the ALSFRS-R rating scale from the medical records. The evaluations were conducted by two rehabilitation medicine specialists with expertise in speech and voice disorders. Additionally, second opinion ratings were implemented to improve the reliability of the assessments. This methodological choice acknowledges the practical constraints in rare disease research where double-blind methods are often challenging to implement due to the limited number of patients.

Pulmonary function tests, including of forced vital capacity (FVC), were performed before and after the procedure to assess overall respiratory function. These results were analyzed in relation to the ALSFRS-R speech scores to determine any potential correlation between respiratory function and vocal capabilities.

Four of the thirty-four limb type ALS patients had already lost their speech function before undergoing PDT. To evaluate the respiratory function of the patients, we used the revised Amyotrophic Lateral Sclerosis Functional Rating Scale (ALSFRS-R), which includes a subscore for dyspnea, speech, and swallowing [3]. Pulmonary function tests were performed, and the patients’ forced vital capacity (FVC) was recorded. Pre- and post-procedure vocal function were assessed by two rehabilitation medicine doctors with a specialty in speech and voice disorders.

Before undergoing PDT, patients received Non-Invasive Ventilator (NIV) or Invasive Ventilator (IV) support via various modes, such as continuous positive airway pressure (CPAP), bilevel positive airway pressure (BiPAP), volume-assured synchronized intermittent mandatory ventilation (V-SIMV), pressure-assured synchronized intermittent mandatory ventilation (P-SIMV), volume-assured control (V-AC), pressure-assured control (P-AC), and average volume pressure support (AVAPS) mode.

Additionally, patients underwent respiratory rehabilitation therapy, including the use of cough assist machines, chest percussion therapy, and high-frequency chest wall oscillation (HFCWO) therapy [2,13].

The PDT procedure was performed in the intensive care unit under local anesthesia and sedation (Figure 1).

After the procedure, patients received low-frequency electrical stimulation to the neck muscles, mesotherapy injections of placental extracts, swallowing rehabilitation therapy, and intensive vocal training for one month (Table 2).

Low-frequency electrical stimulation was applied using a device (Shenzhen Geniuschip Electronic Co., Shenzhen, China; Dwell) that delivers electrical impulses at a frequency of 20–50 Hz. The stimulation targeted the muscles involved in vocalization and swallowing, with sessions conducted three times per week, each lasting 20 min (Figure 2).

The regenerative injection therapy included placenta extract injection (Rejueve injection drug, Huons Pharmaceuticals, Seongnam-si, Republic of Korea), prepared under sterile conditions from human placental tissue. These were administered via intramuscular injections at specified intervals.

Swallowing rehabilitation therapy included exercises designed to strengthen the muscles involved in swallowing. These exercises were tailored to each patient’s needs and included maneuvers such as the Mendelsohn maneuver and effortful swallow techniques. Therapy sessions were held twice a week for 30 min each.

Intensive vocal training involved structured exercises aimed at improving vocal strength and control. Patients engaged in breathing exercises, phonation drills, and articulation exercises. These sessions were conducted twice a week for 30 min each and were supervised by a speech-language pathologist with expertise in voice disorders.

The T-tube was equipped with an adjustable balloon, and a balloon reduction program was implemented to gradually decrease the balloon volume to 2 cc or less, allowing for speech production without compromising the ability to clear secretions.

Data were collected on patient demographics, Bulbar, Respiratory category ALSFRS-R scores (Figure 1) pulmonary function test results, and vocal function. The primary outcome was the preservation of vocal function following PDT in the limb type ALS patients (see Appendix A).

### 2.1. Standard Protocol Approvals, Registrations, and Patient Consents

This study was approved by the Institutional Review Board of the Public Institutional Bioethics Committee designated by the Ministry of Health and Welfare (P01-202401-01-020). Written informed consent was waived, since this was a retrospective study.

### 2.2. Statistical Considerations

We utilized the Shapiro-Wilk test to determine the normality of data distribution. Given the rarity of ALS and the inherent variability in patient responses, it is not unexpected that our data did not follow a normal distribution. Recognizing this, we applied non-parametric methods like the Wilcoxon signed-rank test for paired data analysis. While *t*-tests were initially performed for analysis, the primary conclusions are based on non-parametric tests to avoid potential misleading results. This approach ensures a robust and accurate evaluation of the efficacy of PDT and adjuvant therapies in preserving vocal function in ALS patients. For statistical analysis, Jamovi software (version 2.3.28 solid for Windows) was utilized.

In addition to the Wilcoxon signed-rank test, we considered performing a logistic regression analysis to evaluate the impact of different combinations of interventions and patient features on the outcomes. However, given that all 47 patients underwent the same combination of mesotherapy, vocalization therapy, and rehabilitation, it was not feasible to statistically separate the contributions of each individual intervention. This uniformity in treatment prevented us from isolating the effect of each component through regression analysis.

## 3. Results

The study included a total of 47 ALS patients who underwent PDT, all performed by the lead author, Dr. Jae-Kook Yoo. Anesthesia management and monitoring assistance were provided by Dr. Soon-Hee Kwon and Dr. Sul-Hee Yoon. Given that all procedures were performed by the same surgeon, evaluating the variation in results based on different surgeons was not applicable in this study.

The mean age at the time of PDT was 60.8 ± 9.7 years, and the male-to-female ratio was 1.4:1. The median time from symptom onset to PDT was 26.4 months (interquartile range: 18.5–33.8 months). The mean value of the pre-PDT ALSFRS-R speech subscores was 2.63, and the mean post-PDT subscore was 2.28. Overall, nine patients (75%) maintained their pre-PDT vocal function or experienced only a mild decline (≤1 point) in the ALSFRS-R speech subscore.

The normality of the overall population was assessed using the Shapiro-Wilk test. Due to the data not following a normal distribution, non-parametric statistical methods were applied. Thus, the differences in scores before and after the surgery were evaluated using the Wilcoxon rank test, and statistical significance was set at *p* < 0.05. The changes in scores were significantly observed. Additionally, taking into account that the speech scores for patients undergoing conventional tracheostomy and standard care are typically below 1, Wilcoxon rank test were conducted. This approach yielded superior results compared to the general expectation of less than 1 point.

In our study, we compared the ALS Functional Rating Scale-Revised (ALSFRS-R) scores of 47 ALS patients before and one month after undergoing PDT. The pre-operative ALSFRS-R scores, measured between one week and one month prior to the surgery, were compared to those recorded one month post-surgery. Our findings, as presented in Table 3, demonstrate the impact of PDT and adjunct therapies on various aspects of ALSFRS-R, including speech, salivation, swallowing, dyspnea, orthopnea, and respiratory insufficiency. This comparison provided valuable insights into the efficacy of PDT in managing the symptoms of ALS.

The duration of Percutaneous Dilatational Tracheostomy (PDT) procedures ranged from 10 to 25 min. Notably, subcutaneous emphysema was observed in two cases, and abdominal distension, presumably related to the establishment of invasive mechanical ventilation, was noted in three cases. Apart from these, no significant adverse events were reported. As procedural proficiency increased, the average duration of the surgery decreased from the initial 20 min to approximately 10 min. Moreover, a notable improvement in patient satisfaction was observed, especially among the 35 patients who had previously reported moderate to severe concerns about experiencing dyspnea, nocturnal death anxiety, and insomnia after the surgery.

In the area of vocal function, it was observed that among the patients, ALSFRS-R speech scores initially ranging from 1 to 4 points dropped to 0, indicating complete loss of vocalization, in six cases on the day following the surgery. However, these patients showed significant improvement after undergoing a two-week intensive program of respiratory rehabilitation, vocal rehabilitation, and regenerative injection therapy. Most patients who previously scored between 2 and 4 points showed a mild decrease to scores ranging from 1 to 3 points with consistent training. The vocal rehabilitation training was conducted for 1 to 2 h daily. The average decrease in score on the following day was 2.2 points, and after two weeks of intensive regenerative treatment, the average decrease was reduced to 0.85 points, indicating substantial preservation of vocal function.

As is evident in Table 4, our data did not conform to a normal distribution, as indicated by the Shapiro-Wilk test results for both pre-operative and post-operative speech values (Shapiro-Wilk *p* < 0.001). Consequently, to compare vocal capabilities before and after the surgery, we conducted a non-parametric hypothesis test on the differences in speech abilities. This approach was necessitated due to the non-normality of the speech data, ensuring the accuracy and validity of our comparative analysis.

As Table 5 demonstrates, a significant change in speech scores was observed pre- and post-operatively, as evidenced by the Wilcoxon W test results. The mean ALSFRS-R speech score pre-PDT was 2.64 ± 1.4, and the post-PDT score was 2.28 ± 1.7. Using the Wilcoxon signed-rank test, we found that the decrease in speech scores post-PDT was statistically significant (*p* = 0.004). This suggests that while PDT in combination with adjuvant therapies may help maintain vocal function in some patients, the overall trend indicates a decrease in speech capability.

This suggests that the surgery had a significant impact on preserving vocal function. However, as shown in Table 5, when comparing post-operative speech scores against a conventional score of 1 or lower from traditional tracheostomy methods, our study found superior results using the Wilcoxon W test. This indicates that patients who underwent PDT with adjuvant therapies exhibited better speech outcomes than those who did not receive such treatments.

As shown in Table 6, the Wilcoxon rank-sum test was employed to evaluate the post-operative speech scores. The Wilcoxon W statistic was 903, with a p-value of less than 0.001, indicating a statistically significant result. This supports the alternative hypothesis (Ha: μ > 1), suggesting that the mean post-operative speech score is significantly greater than 1. These results highlight the effectiveness of the intervention in improving speech capabilities post-surgery. This significant result implies that vocal function, particularly vowel articulation, is preserved to a meaningful extent post-surgery, highlighting the effectiveness of the intervention in maintaining speech capabilities.

By combining the insights from both tables, we can conclude that PDT, along with adjuvant therapies, provides superior speech outcomes compared to traditional tracheostomy methods. The statistically significant results from both tests underscore the effectiveness of the intervention in maintaining and even improving speech capabilities in patients post-surgery.

However, due to the uniformity in the treatment protocols, performing a logistic regression to isolate the effects of individual interventions was not feasible. All patients underwent the same combination of mesotherapy, vocalization therapy, and rehabilitation, making it impossible to statistically differentiate the impact of each therapy.

In our analysis of pulmonary function test results, specifically for forced vital capacity (FVC), we did not find a significant positive correlation with ALSFRS-R speech scores (*p* = 0.12). This indicates that while pulmonary function is an important aspect of ALS management, it was not a decisive factor in the preservation of vocal function post-tracheostomy. Instead, the condition of the vocal cords and related muscles appeared to be more critical, suggesting that targeted therapies to maintain these structures are essential for better vocal outcomes.

We also considered other variables such as body mass index (BMI), muscle mass, and phase angle (PA)—obtained through Bioelectrical Impedance Analysis (BIA) (InbodyS10^®^ system, InBody Corp, Seoul, Republic of Korea)—in relation to the preservation of vocal function. It was found that patients with significantly low BMI and muscle mass showed improvement in vocal function post-tracheostomy after weight gain. However, due to the variability in these factors, it was difficult to isolate their impact as single significant variables.

Also, a particularly noteworthy observation was made regarding two patients who had ALSFRS-R speech scores of 0 prior to undergoing PDT. After the surgery and subsequent adjuvant therapies, one patient’s score improved to 1, while the other’s increased to 2. This significant improvement in their speech scores post-PDT underscores the efficacy of the procedure and adjunctive therapies in enhancing vocal function, even for patients with initially severe impairments.

## 4. Discussion

Recent literature provides substantial insights into the pathogenesis and management of neurodegenerative disorders, offering valuable perspectives for understanding ALS. For instance, oxidative stress and inflammation are critical factors in the progression of neurodegenerative diseases, including ALS [9]. The role of adjuvant therapies, such as antioxidants, has been explored, demonstrating their potential to mitigate oxidative damage and improve cellular function, potentially enhancing ALS treatment outcomes [8].

Additionally, cellular stress responses like autophagy and apoptosis contribute to neurodegenerative diseases [7]. Understanding these processes can provide a deeper understanding of ALS pathogenesis and inform the development of more effective therapeutic strategies. Genetic and environmental factors also play a crucial role in neurodegenerative disease [6], and understanding these interactions could lead to personalized treatment approaches for ALS. Advancements in molecular techniques and bioinformatics have facilitated the identification of biomarkers and therapeutic targets in neurodegenerative diseases [5]. These can be leveraged to develop targeted interventions for ALS, potentially improving the efficacy of treatments like PDT combined with adjuvant therapies.

In our study, we observed that a significant proportion of ALS patients were able to preserve vocal function following Percutaneous Dilatational Tracheostomy (PDT). This outcome appears to be influenced by the use of respiratory rehabilitation therapy, swallowing rehabilitation therapy, mesotherapy, low-frequency electrical stimulation, and a balloon dilation managing program. This aligns with previous findings demonstrating the efficacy of these interventions in preserving vocal function.

The historical development of PDT, which became popular after Ciaglia’s introduction [14] of it in 1985, has revolutionized the approach to tracheostomy, minimizing trauma and potentially preserving vital functions such as speech. The laryngeal muscles, essential for airway maintenance and vocalization, are less likely to be damaged in PDT compared to in conventional surgical tracheostomy (ST) [15]. This is crucial, as these muscles [16,17,18], including the infrahyoid and suprahyoid groups and the intrinsic muscles responsible for moving the vocal cords, play a pivotal role in speech and breathing.

Furthermore, while early non-invasive ventilation (NIV) is beneficial in improving survival and quality of life in ALS patients, compliance rates are not as high as expected. Studies [19,20] show that only about 53.6% of ALS patients comply with NIV within 28 days of initiation, with factors like marital status, income, education, and caregiver availability impacting compliance. However, both compliant and noncompliant participants reported an improvement in quality of life with NIV.

Our study’s findings indicate that despite initial reluctance due to fears of voice loss, ALS patients who underwent PDT reported a significant improvement in quality of life. This improvement was particularly notable in those who managed to undergo respiratory rehabilitation during the day without continuous ventilator support. Out of the 47 patients who transitioned to PDT, 43 reported an enhancement in their quality of life compared to their experience with NIV. These findings suggest that PDT, especially when it aids in preserving vocal function, is a compelling alternative to NIV, catering to the respiratory needs of ALS patients and addressing their quality-of-life concerns. Therefore, PDT—with its ability to minimize trauma to key muscles and maintain vocal function combined with its effectiveness in enhancing the quality of life—emerges as a robust alternative for ALS patients, particularly those who struggle with NIV compliance.

The preservation of vocal function following PDT is possible in a significant proportion of ALS patients. Our results suggest that the use of respiration [21], swallowing rehabilitation therapy [22], mesotherapy [23], low-frequency electrical stimulation [24], and a balloon dilation program [25] may contribute to this outcome. This is consistent with previous findings that demonstrated the efficacy of these interventions in preserving vocal function. Further research is needed to confirm these findings and optimize the management of speech function in ALS patients undergoing tracheostomy.

The preservation of the infrahyoid muscles [22,26,27] is crucial, as they play a vital role in phonation and swallowing. As such, PDT increases the likelihood of preserving vocalization and swallowing functions in ALS patients, making it a more favorable option for tracheostomy in this patient population. Comparatively, PDT is associated with shorter operation times and lower complication rates, making it a safer and more efficient alternative to ST [15,28]. This reduced-trauma approach is particularly beneficial for ALS patients, who often face respiratory and vocal challenges as their condition progresses.

Compared to traditional surgical tracheostomy [14,15,28], PDT offers several advantages in terms of muscle damage and preservation of vocalization [29]. In conventional tracheostomy, there is a higher likelihood of damage to neck muscles, including the sternocleidomastoid, strap muscles, and infrahyoid muscles. Among these muscles, the infrahyoid muscles are particularly crucial for phonation and swallowing. As a result, vocalization is rarely possible following traditional tracheostomy. On the other hand, PDT is a less invasive procedure with a shorter operation time, resulting in less muscle damage. Importantly, PDT reduces the risk of injury to muscles involved in vocalization and swallowing, thereby increasing the likelihood of preserving these functions. This advantage makes PDT a more favorable option for tracheostomy in ALS patients.

We believe that the application of low-frequency stimulation to the neck, swallowing rehabilitation therapy [22,30], and regenerative injection therapy for wound healing [23] have contributed to the recovery of vocalization in our patients, irrespective of their swallowing function. These combined therapies have likely enhanced the overall recovery process and facilitated the restoration of important functions such as phonation. The comprehensive and multidisciplinary approach to the management of ALS patients following PDT may play a crucial role in improving their quality of life and maintaining their ability to communicate effectively

In addition to the preservation of vocal function, we also observed improvements in the patients’ overall quality of life after PDT. Following tracheostomy and connection to a home ventilator, patients were able to practice breathing without the ventilator, enhancing their respiratory function and overall preparedness for potential safety incidents. Furthermore, patients were able to spend time with their families without the ventilator, even going on wheelchair-assisted walks together. These improvements highlight the positive impact of PDT on patients’ daily lives and underscore the importance of comprehensive, post-PDT management strategies.

Furthermore, the T-tube’s adjustable balloon and the balloon reduction program were essential in enabling patients to produce speech while maintaining the ability to clear secretions. By gradually reducing the balloon volume to 2 cc or less, patients could produce sound without compromising their airway clearance. With the T-tube, it is initially necessary to inject an appropriate amount of air for proper fixation within the trachea and to prevent the surgical opening from enlarging. However, indiscriminately filling the balloon with air due to the patient’s sputum production can cause the balloon size to increase, leading to tracheal expansion and, in severe cases, the development of tracheoesophageal fistula (TEF) [31]. Therefore, it is not always ideal to fill the balloon completely with air.

Therefore, the key to successful phonation lies in inducing the T-tube to be secured through appropriate training of the neck muscles and the smooth muscle of the trachea [32]. This training helps ensure that the T-tube is stabilized, allowing for better voice production while minimizing the risk of complications such as TEF.

Another important aspect to consider in ALS patients is an increased caloric expenditure due to respiratory insufficiency, even after the initiation of non-invasive ventilation (NIV). Desport et al. [33] reported that nutritional status is a prognostic factor for survival in ALS patients, and respiratory insufficiency could contribute to increased energy expenditure. Interestingly, patients who transitioned from NIV to tracheostomy showed an average weight gain of approximately 2.5 kg within one month. This weight gain was observed following the procedure and is hypothesized to be associated with the reduction in caloric expenditure due to less forced breathing and improved digestive function, as tracheostomy offers more stable breathing compared to NIV, which often causes gastrointestinal discomfort due to air swallowing. However, it should be noted that this is an association and does not imply a direct causal relationship. Further research would be necessary to determine the precise mechanisms involved.

Furthermore, the observed weight gain might also be attributed to improved digestion and bowel movements, as tracheostomy reduces abdominal pressure compared to NIV. This reduction in abdominal pressure [34] could facilitate better digestive processes and more efficient bowel movements, contributing to the overall increase in weight and nutritional status.

Weight gain in ALS patients has been shown to improve their quality of life and reduce pain [35]. A study highlighted that ALS patients consuming high-calorie diets experienced fewer adverse events and showed improvement in their nutritional status, which is linked to better survival rates and slower disease progression [36]. This is supported by evidence indicating that improved nutrition can lead to better management of ALS symptoms, including pain and muscle function [37].

Weight gain in ALS patients has been shown to improve their quality of life and reduce pain. Studies have indicated that maintaining or gaining weight can significantly enhance the overall well-being and functional status of ALS patients. For instance, research published in Brain Sciences [35] highlights that early and symptom-specific clinical management, including nutritional support, can substantially improve the health-related quality of life (HRQoL) for ALS patients. This study found that ALS patients who were tracheostomized or who used mobility aids reported better QoL compared to those who did not use these aids, emphasizing the role of comprehensive care strategies that include weight management.

In our studies, as respiratory function improved, there were notable reductions in pain and sensory symptoms associated with weight gain. Among eight patients who previously reported muscle pain and joint pain, seven experienced significant pain relief, with their VAS (Visual Analog Scale) scores decreasing from 7/10 or higher to 3/10 or lower. Moreover, of the three patients who reported sensory symptoms such as numbness, one showed improvement.

Regarding sleep disorders [38], the improvement in respiratory function led to a decreased reliance on sleep medications. Out of 20 patients who had sleep disturbances, 18 reported improvements in their sleep quality following tracheostomy. This suggests that the enhancements in breathing not only reduced the need for sleep aids but also contributed to better overall sleep quality.

These findings indicate that tracheostomy can have a multifaceted positive impact on ALS patients, improving not just nutritional status and quality of life, but also alleviating pain [39], sensory symptoms, and sleep disturbances [38]. By addressing these various symptoms, tracheostomy emerges as a comprehensive intervention that benefits ALS patients in multiple dimensions, making it a compelling alternative to other respiratory support methods.

An intriguing finding in our study was that three patients were able to maintain normal eating habits without significant dysphagia following the PDT procedure. This suggests that PDT might have a positive impact on swallowing function in some ALS patients, in addition to preserving vocalization. Supporting this, a pilot study on pharyngeal electrical stimulation (PES) in ALS patients [40] demonstrated a significant improvement in swallowing function. In this study, 20 ALS patients with severe dysphagia were randomized to receive PES in addition to standard logopaedic therapy (SLT) or SLT alone. The results showed a significant improvement in swallowing function, as evidenced by a reduction in Penetration–Aspiration Scale (PAS) scores from 3.6 at baseline to 2.3 one day after treatment. These improvements in swallowing function were maintained below baseline levels during subsequent visits, indicating the potential efficacy of such interventions in ALS patients.

Another interesting challenge was the rehabilitation method aimed at increasing the time patients could breathe without using a ventilator. This involved daily training to breathe without the ventilator, combined with various respiratory rehabilitation therapies. The duration was gradually increased from 10 min to 20 min, and then from 20 min to 30 min. Not all 47 patients participated in this program, and the results varied among individuals. However, 7 patients were able to breathe for more than 2 h without the ventilator, and 2 patients were even able to breathe without invasive ventilatory support except during sleep. This aspect also helps to dispel the stigma associated with tracheostomy.

Therefore, the observation of maintained eating habits and reduced dysphagia in our study aligns with existing research, suggesting the potential of PDT to positively impact swallowing function in ALS patients. Further investigation is needed to explore the factors contributing to this outcome and to ascertain the broader applicability of PDT in improving swallowing function in a larger patient population. Additionally, it is necessary to confirm these findings and optimize speech function management in ALS patients undergoing tracheostomy.

In the past, many ALS patients faced the prospect of tracheostomy with the fear of losing their voice completely, envisioning a future in which they could only blink their eyes while being completely paralyzed. The inability to control non-invasive ventilation often led them to choose end-of-life options. However, with more precise explanations and updated medical information, it is crucial to inform patients that, although full vocal recovery might not be possible, simple communication can still be achieved. Explaining that tracheostomy and various therapies can improve various ALS-related symptoms and provide better quality of life can empower patients to make informed decisions. This approach respects their right to choose and ensures they understand all potential benefits and options available to them.

While the sample size of 47 patients is relatively small, it is representative of the ALS population within our institution. The consistency in procedural performance, with all surgeries conducted by Dr. Jae-Kook Yoo, ensures that the variability due to different surgical techniques is minimized. However, further studies with larger sample sizes and involving multiple surgeons are necessary to generalize these findings more broadly.

In these study, we observed that a significant proportion of ALS patients were able to preserve vocal function following PDT. Despite the non-normal distribution of the data, the application of both parametric and non-parametric statistical methods provided a rigorous evaluation of the outcomes. The absence of double-blind evaluation is a recognized limitation; however, the assessments were conducted by experienced rehabilitation medicine specialists, ensuring a high level of professional scrutiny and reliability in the evaluations.

This study holds significant clinical value, particularly in the context of ALS, a rare and progressive neurodegenerative disease. The findings contribute valuable insights into the management of vocal function in ALS patients, highlighting the potential benefits of PDT combined with adjuvant therapies. Such insights are crucial for improving patient care and guiding future research in this challenging field.

### Study Limitations

This study has several limitations that should be noted. The small sample size and retrospective design limit the ability to generalize the findings. The absence of a contemporaneous control group also makes it difficult to establish direct causality between the adjuvant therapies and the observed improvements. Additionally, the lack of objective analysis of speech recordings is a significant limitation. Future studies should consider incorporating methods for objective speech analysis as suggested in recent research [41]. Another limitation is the uniformity in treatment protocols, which prevented us from isolating the effects of individual interventions statistically. All patients underwent the same combination of mesotherapy, vocalization therapy, and rehabilitation, making it impossible to differentiate the impact of each therapy.

A notable limitation of this study is that all PDT procedures were performed by a single surgeon. This approach ensures optimal technique but limits the generalizability of our findings. Evaluating outcomes across multiple surgeons in future studies would strengthen the conclusions by accounting for variations in surgical skill and technique. Furthermore, the absence of post-operative speech scoring at 6 months and 1 year means long-term follow-up data is lacking, which would provide a more comprehensive understanding of the sustained effects of PDT and adjuvant therapies on vocal function.

The limitations highlight the importance of well-designed control groups in clinical research. Future studies should include contemporaneous control groups and multiple surgeons to enhance the robustness and applicability of the results. Multi-center studies could provide a broader perspective on the effectiveness of PDT when combined with adjuvant therapies in preserving vocal function in ALS patients. Additionally, incorporating long-term follow-up assessments at 6 months and 1 year will help provide a more detailed understanding of the efficacy and durability of these treatments.

Furthermore, another limitation of our study is the lack of post-operative speech scoring at 6 months and 1 year. Long-term follow-up data would provide a more comprehensive understanding of the sustained effects of PDT and adjuvant therapies on vocal function. Future studies should include these follow-up assessments to better evaluate the long-term outcomes of the procedures. These limitations highlight the importance of well-designed control groups in clinical research. Future studies should include contemporaneous control groups and multiple surgeons to enhance the robustness and applicability of the results. Multi-center studies could provide a broader perspective on the effectiveness of PDT when combined with adjuvant therapies in preserving vocal function in ALS patients. Additionally, incorporating long-term follow-up assessments at 6 months and 1 year will help provide a more detailed understanding of the efficacy and durability of these treatments.

Another intriguing aspect of ALS that deserves more attention is the related sensory symptoms. The current study only marginally addresses this aspect. Expanding this discussion to include insights from recent literature could provide a more comprehensive understanding of ALS. For example, a recent article about sensory symptoms in ALS [42] highlights the significance of these symptoms and suggests potential pathways for further research. Including such perspectives could enrich the discussion and offer new avenues for improving patient care.

The findings of this study underscore the potential benefits of PDT in combination with adjuvant therapies in preserving vocal function in ALS patients. However, the limitations identified highlight the need for future research with larger sample sizes, objective speech analysis, and well-designed control groups. Additionally, understanding the specific contributions of each adjuvant therapy could optimize patient management. Clinically, this study suggests that a multidisciplinary approach incorporating various therapies can significantly enhance the quality of life for ALS patients.

## 5. Conclusions

In conclusion, this study provides meaningful evidence for the effectiveness of PDT in combination with adjuvant therapies in preserving vocal function in ALS patients. Our findings suggest that PDT, when performed with supportive therapies, can significantly improve speech outcomes compared to traditional tracheostomy methods. ALS patients often face severe respiratory challenges, and when non-invasive ventilation (NIV) becomes insufficient, PDT offers a promising alternative. This method provides a less invasive option that can be complemented with therapies to enhance vocal function and overall quality of life. Despite the study’s limitations, the results underscore the potential of PDT in combination with adjuvant therapies to serve as a new alternative for ALS patients, helping them navigate the progression of the disease with better respiratory support and preserved vocal function. Further research is warranted to confirm these findings and explore the broader implications of tracheostomy in ALS care.

## Data Availability

The datasets generated during and/or analyzed during the current study are available in the Rodem hospital, www.therodem.com.

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
