# Peer review of "Preservation of Vocal Function in Amyotrophic Lateral Sclerosis (ALS) Patients Following Percutaneous Dilatational Tracheostomy (PDT) and Adjuvant Therapies"

_biomedicines, 2024, doi:10.3390/biomedicines12081734_

Round 1
Reviewer 1 Report (Previous Reviewer 4)
Comments and Suggestions for Authors
The authors present a study examining preservation of vocal function in Amyotrophic Lateral Sclerosis patients following percutaneous dilatational tracheostomy (PDT). The sample size is small but sufficient for an exploratory study.
The revisions have improved the technical quality of the paper - particularly, the authors have addressed concerns with the presentation of quantitative results, including suggested changes to the statistical analysis.
The overall presentation is much improved and the tone has been adjusted to better reflect the interpretation of the results. Only one result might still be overstated - the result that there was a significant association with weight gain following the procedure. The authors should state this as an association versus imply a direct causal relationship given causality cannot be directly proven in this study. The authors hypothesis that weight gain is due to less caloric expenditure is fine to state, but the interpretation should not be stated as an absolute fact.
Author Response
Thank you for your comments

Reviewer 2 Report (New Reviewer)
Comments and Suggestions for Authors
First of all I would like to congratulkate the authors on this study which targets patients with an agressive neurodegenerative disease, Amyotrophic lateral sclerosis (ALS), with a very bad prognosis in time. As the disease advances the patients require respiratory assistance which is provided gradually as the disease worsens. They study tries to establish, based on the experience of the clinic the influence of PDT and other adjuvant therapies used to suport respiration on the preservation of the vocal function.
The study is done on a rare disease, which justifies the low population in the statistical analysis.
While the paper is hard to read because of the track changes which are present in the pdf file, the quality of the work si very high based on my opinion.
All the data is well presented and I would just recommend a re-reading of the paper to correct some minor language mistakes.
While I am not very familiar with this disease, I have read the paper several times and to my knowledge the entire patient management is well done and well presented. (I've mentioned my limited understanding of ALS as not being able to decide whether the most efficient medical protocols were applied).
With the addition of the supplementary material, I am confident to say that this is a valuable study and should be published.
Please be careful next time with the track changes as it is very annoying to read the paper in this way.
Comments on the Quality of English LanguageThe language is fine I spotted just several minor mistakes, BUT, a careful re-reading should be done after the removal of the track changes.
Author Response
Thank you for your comments.

Reviewer 3 Report (New Reviewer)
Comments and Suggestions for Authors
This work examined the effectiveness of percutaneous dilated tracheostomy (PDT) combined with adjunctive therapies in preserving vocal function in patients with amyotrophic lateral sclerosis (ALS). The work provides important contributions for the management of ALS, especially in relation to the quality of life of patients.
The title is clear and effectively describes the main objective of the study. However, it could be made more concise.
Overall the work is well written and reads well. As with the title, some paragraphs are too long, so I suggest dividing them into shorter, more concise sentences to improve readability.
It would be helpful to include a more detailed description of patient inclusion and exclusion criteria and data collection procedures.
Finally, I suggest including a “study limitations” section.
A positive correlation was found between additional therapies and preservation of vocal function, with significant improvements in patients' quality of life. However, not all patients benefited equally: the results show that 36 of the 47 patients maintained or slightly improved their language scores post-PDT. This may be due to the small sample size and lack of a control group making it difficult to establish direct causality between the additional therapies and the observed improvements. Furthermore, it is unclear what the specific contribution of each adjuvant therapy is within the combined approach. All this, together with the retrospective design, represents a limitation of the study. Another limitation is the lack of objective analysis speech recording, please see a recent article on the topic (https://doi.org/10.1016/j.bspc.2023.105706) and comment.
The sensory symptoms represent another intriguing aspect of the disease, that the authors have discussed only marginally. I would recommend expanding this section in the discussion, including a recent article about sensory symptoms in ALS (10.1093/brain/awad426).
In conclusion, the integration of various therapies and the multidisciplinary approach of this work can offer significant benefits to patients in the management of ALS. This is a significant strength. This work highlights the importance of improving patients' quality of life, a crucial aspect in the treatment of ALS.
Author Response
Thank you for your comments.

This manuscript is a resubmission of an earlier submission. The following is a list of the peer review reports and author responses from that submission.
Round 1
Reviewer 1 Report
Comments and Suggestions for Authors
Thank you for the opportunity to review this manuscript.
I value the thorough presentation of the PDT treatment and its clinical application, outcomes, and conglomerate treatments in the discussion section of the paper. The background section is, however, sparse in details to say the least. The authors should restructure their manuscript.
The speech evaluation is based on a retrospective evaluation of the ALSFRS-R rating scale from medical records. There was no blinding in the evaluation of patient symptoms according to this scale, and the authors do not report on who performed the evaluation (professional status, experience, second opinion ratings, and so on). Consequently, the level of evidence of an improvement following PDT and the other therapies given by the report is low.
The authors conclude repetetly that the data were not normally distributed, but it is difficult to understand why they would assume that an ordinal rating scale with few rating responses would result in a normally distributed response. The observation of non-normality does not prevent the authors from performing t-tests, which I cannot see as anything other than a high risk of misleading results being presented.
The authors report that pulmonary tests were performed and forced vital capacity being recorded, but these are not reported on fully and are not connected with the outcome of the speech evaluation, and I cannot see why.
Reviewer 2 Report
Comments and Suggestions for Authors
Yoo et al demonstrated that percutaneous dilatational tracheostomy (PDT) can be a viable option for preserving vocalization in ALS patients, particularly those with limb onset disease. This is an interesting paper, showing that the majority of treated patients were able to maintain their voice, which significantly improves their quality of life and communication abilities. There are, however, several issues to be addressed to further improve the manuscript/
1. Placental extracts need to be specified.
2. Specifics on low frequency electrostimulation, swallowing rehabilitation therapy, and intensive vocal training need to be provided.
3. More cases should be desirable. What is the degree of variation in results from surgeon to surgeon?
Reviewer 3 Report
Comments and Suggestions for Authors
This is a very misleading manuscript that must be rejected. The abstract gave completely wrong interpretation of the results. In the results section, the authors reported notably decreased speech scores post-PDT compared to pre-PDT, indicating that PDT has a negative impact on the speech capability. However, the abstract stated that patients maintained or even showed slightly improved speech capability post PDT. The “one sample” T-test used to compare patients receiving PDT and traditional tracheostomy is also not acceptable, as the sample size and mean scores of the patients receiving traditional tracheostomy are not available.
Reviewer 4 Report
Comments and Suggestions for Authors
The authors examine the impact of adjuvant combination therapies for the preservation of vocalization after percutaneous dilational tracheostomy in ALS patients. Results illustrate the interventions are significantly improved over baseline. The overall paper is very good, and I have only a couple of suggestions for further improvement.
The data analysis is correct as presented. However, it would be interesting to see if the authors could deduce further signal to see what specific combinations of interventions and/or patient features are most driving the significant change. One suggestion might be to perform a logistic regression and evaluate the coefficients or to examine feature weights. While they might not be significant due to sample size, it may provide additional valuable insight. The other less appealing option would be to divide the data into sub-groups and perform a hypothesis test with a correction factor for multiple comparisons.
Finally, a figure or table that includes the full list of adjuvant therapies would be helpful to better encapsulate the comprehensiveness of the study approach.
MINOR: Where possible, please reduce first person usage in the text (our, we, etc.).
